# Investigation of Microstructure and Mechanical Properties of SAC105 Solders with Sb, In, Ni, and Bi Additions

**DOI:** 10.3390/ma16114059

**Published:** 2023-05-30

**Authors:** Yaxin Gao, Xilei Bian, Xingbao Qiu, Yandong Jia, Jun Yi, Gang Wang

**Affiliations:** 1Institute of Materials, Shanghai University, Shanghai 200444, China; gyx@shu.edu.cn (Y.G.); qiuxingbao@shu.edu.cn (X.Q.); yandongjia@shu.edu.cn (Y.J.); jxy305@shu.edu.cn (J.Y.); g.wang@shu.edu.cn (G.W.); 2Zhejiang Institute of Advanced Materials, Shanghai University, Jiashan 314100, China

**Keywords:** micro-alloying, microstructure, mechanical properties, SAC105 solder

## Abstract

Low Ag lead-free Sn-Ag-Cu (SAC) solders have attracted great interest due to their good drop resistance, high welding reliability, and low melting point. However, low Ag may lead to the degradation of the mechanical properties. Micro-alloying is an effective approach to improving the properties of SAC alloys. In this paper, the effects of minor additions of Sb, In, Ni, and Bi on microstructure, thermal and mechanical properties of Sn-1 wt.%Ag-0.5 wt.%Cu (SAC105) were systematically investigated. It is found that the microstructure can be refined with intermetallic compounds (IMCs) distributed more evenly in the Sn matrix with additions of Sb, In, and Ni, which brings a combined strengthening mechanism, i.e., solid solution strengthening and precipitation strengthening, leading to the tensile strength improved of SAC105. When Ni is substituted by Bi, the tensile strength is further enhanced with a considerable tensile ductility higher than 25%, which still meets the practical demands. At the same time, the melting point is reduced, the wettability is improved, and the creep resistance is enhanced. Among all the investigated solders, SAC105-2Sb-4.4In-0.3Bi alloy possesses the optimized properties, i.e., the lowest melting point, the best wettability, and the highest creep resistance at room temperature, implying that element alloying plays a vital role in improving the performance of SAC105 solders.

## 1. Introduction

Traditional tin–lead (Sn-Pb) solders have been widely used in the electronics industry due to their low cost, and excellent physical, thermal, and mechanical properties [1,2]. However, Pb was forbidden by RoHS Law for its harmful to the environment and human health [3]. Therefore, the dismission of the toxicity of lead stimulates the search for lead-free alternatives in electronic applications. The eutectic Sn-Ag-Cu (SAC) lead-free solders, such as Sn-3 wt.%Ag-0.5 wt.%Cu (SAC305) [4], Sn-3.8 wt.%Ag-0.7 wt.%Cu (SAC387) [5], and Sn-3.9 wt.%Ag-0.6 wt.%Cu (SAC396) [6], considered as a substitute for Sn-Pb solder due to their low melting temperature, high tensile strength, and good thermo-mechanical fatigue properties. Nevertheless, many studies have demonstrated that these SAC alloys with high Ag amounts usually have low elongation due to the formation of hard and brittle intermetallic compounds (IMCs), which are prone to failure under high strain rates and large temperature ranges of drop impact situations [7,8,9]. Recently, SAC solders with a low Ag content have been identified as promising candidates to replace the traditional Sn-Pb solder [10]. The Sn-1 wt.%Ag-0.5 wt.%Cu (SAC105) is an attractive low-Ag solder alloy in the SAC family because of its good drop failure resistance in electronic interconnects [11,12,13]. However, the decrease in Ag brings problems such as low strength, poor creep resistance, and poor wettability [14,15,16]. Considerable efforts have been devoted to developing SAC105-based solders with enhanced properties.

Doping with alloying elements has been proven to be an effective and practical approach to optimizing the performance of SAC solder alloys [17]. With an aim to obtain SAC solders with high strength, good creep resistance, good wettability, and low melting point, various elements have been micro-alloyed and explored [18,19,20,21]. Among them, the alloying elements such as antimony (Sb), indium (In), nickel (Ni), and bismuth (Bi) are effective in the modulation of the microstructures and properties of SAC solder alloys. Sb can improve the strength of SAC solder alloys. For example, the tensile strength of Sn-3.5 wt.%Ag-0.7 wt.%Cu-*x*Sb alloy can be increased by 15% as the content of Sb increases from 1 wt.%-2 wt.% since Sb can replace Sn in the sublattice with Cu_6_(Sn, Sb)_5_ and Ag_3_(Sn, Sb) to form solid solutions that operated with strengthening effects [22,23]. Moreover, it was found that adding Sb could reduce the size of Cu_6_Sn_5_ precipitations and the thickness of the intermetallic layer of SAC solders, improving the quality of solder joints [24]. Due to the low melting point of In (156.9 °C), Hung et al. developed an Sn-Ag-based lead-free solder with a low melting temperature by adding a small amount of In [25]. Moreover, In can lower the melting temperature, and the addition of In can also help optimize the performance of an Sn-0.3 wt.%Ag-0.7 wt.%Cu alloy the solder alloy through the decrease in the average grain size [26]. Ni can optimize the microstructure of solders [27,28]. Adding 0.1 wt.% Ni to Sn-3.5 Ag can greatly inhibit the growth of Ag_3_Sn during welding and subsequent solid-state aging processes [29,30]. Hammad reported that Ni elements could diffuse from the molten solder matrix into the intermetallic compound particles, forming Ni_3_Sn_4_ IMCs during the solidification process, thus improving the microstructure of solders and increasing the life of electronic components [31]. Bi can improve strength and wettability. With the addition of 1 wt.%Bi to SAC105, the size of Ag_3_Sn and Cu_6_Sn_5_ IMCs is decreased, and the yield stress and tensile strength are increased due to the solution strengthening mechanism of Bi [32]. Moreover, Bi can not only inhibit the degradation of microstructure and mechanical properties during aging but also can improve the wettability and reduce the melting temperature of Sn-based solder to a more acceptable level [33,34,35]. However, excessive addition of Bi will cause the solders to be stronger but more brittle, which will accelerate the failure of the solders [36,37,38].

Even though some studies have focused on the effects of Sb, In, Ni, and Bi on the microstructure and properties of SAC alloys, most of them solely took the effect of adding single or two alloying elements into account. However, the simultaneous impact of additions of Sb, In, Ni, and Bi on the microstructure and properties of SAC105 alloys is still missing. In this study, the impact of Sb, In, Ni, and Bi additions on the microstructure, mechanical properties, creep resistance, melting properties, and wettability of low-Ag SAC105 solders were systematically investigated. The mechanism of the influence of alloying elements on the properties of SAC105 was discussed in detail. Accordingly, the optimized composition of SAC105-2Sb-4.4In-0.3Bi solder with good comprehensive properties was attained.

## 2. Experimental Methods

### 2.1. Processing of Lead-Free Solder

SAC solders with nominal chemical compositions (Table 1) were prepared by melting mixtures of high-purity (above 99.99%) metals of Sn, Ag, Cu, Sb, In, Ni, and Bi. Alloy ingots were melted in a vacuum induction furnace (KYKY WK II vacuum induction melting furnace) at 800 °C for 1 h under high purity argon atmosphere and followed by pouring the molten metal into a steel mold to prepare the alloy samples with a size of 50 × 20 × 10 mm^3^.

### 2.2. Mechanical Properties

The tensile properties were tested by a mechanical testing machine (CMT-5205, SANS, Shenzhen, China) at room temperature at a constant strain rate of 3 × 10^−3^ s^−1^. As shown in Figure 1, dog-bone-shaped tensile specimens with a gauge dimension of 12 × 3 × 2 mm^3^ were cut from the as-cast ingots. In this paper, the effects of the addition of Sb, In, Ni, and Bi on the tensile strength, yield strength, and elongation of solder alloys were studied. To ensure repeatability, at least 3 specimens were tested for each alloy component. The tensile fracture morphologies were examined by scanning electron microscope (SEM, Zeiss Sigma 500, Jena, Germany).

### 2.3. Characterization of Physical Properties

#### 2.3.1. Microstructure Characterization

Phase analysis was performed by X-ray diffraction (XRD, Rigaku/Ultima IV) using Cu-Kα radiation with an accelerating voltage of 40 kV and a scanning step of 4 °/min in the angle range of 20°–80°. The microstructure of the solder alloys was characterized by SEM equipped with energy-dispersive X-ray spectroscopy (EDS, Bruker, Bremen, Germany).

#### 2.3.2. Melting Properties

The melting and solidification temperatures of the solder alloys were measured by differential scanning calorimetry DSC (Netzsch TA). For each solder, the sample weighing 15–30 mg was placed in an alumina crucible under argon atmosphere. The sample was first heated to 260 °C with a heating rate of 5 °C/min and then cooled to room temperature at the same rate.

#### 2.3.3. Wettability Test

The wettability of the solders was conducted by a high-temperature wettability angle tester (OCA25HTV, Data Physics Instruments, Stuttgart, Germany), during which the temperature was increased from room temperature to 500 °C at a heating rate of 5 °C/min with a holding time of 1 h, and then decreased to room temperature with the same rate of 5 °C/min. In the wettability test, the mass of the solder ball was 0.3 g, and the size of the copper plate used as the substrate was 3 × 3 × 3 mm^3^. The solder flux of RMA218 from Seamark ZM was used for the wettability test. The wettability tests were performed three times for each alloy composition.

### 2.4. Creep Resistance

Creep tests were carried out using a dynamic thermo-mechanical analyzer (Discovery DMA850, TA instruments, New Castle, DE, USA) under two conditions: (a) with constant stress of 12 MPa at temperatures of 75 °C, 100 °C, 125 °C, and 150 °C, respectively; (b) with different applied stresses of 12, 14, 16, and 18 MPa at room temperature. All solder samples were cut into strips with a dimension of 50 × 3.3 × 0.5 mm^3^ (length× width× thickness) and then were polished to ensure a smooth surface for creep testing.

## 3. Results and Discussion

### 3.1. Mechanical Property

The room temperature stress–strain curves of the solder alloys are shown in Figure 2. From this, the ultimate tensile strength (UTS), yield strength (YS), and total elongation can be obtained, which are summarized in Table 2. It can be seen that SAC105 shows the lowest YS (13.8 MPa) and UTS (22.7 MPa) but excellent elongation (47.1%). By the addition of 2 wt.% Sb to SAC105 can improve the tensile strength of the solder, this is because Sb can form solid solutions in Sn, Cu_6_Sn_5_, and Ag_3_Sn phases, and can reduce the size of Cu_6_Sn_5_ precipitates and the thickness of the intermetallic layer, resulting in an increase in strength even though a slight decrease in elongation [39]. Adding 4.4 wt.% In to SAC105 can greatly improve the strength of the solder, which is almost twice stronger as compared with SAC105. Meanwhile, due to the adverse effect of coarse IMCs on the performance of the solder, the elongation is greatly reduced. In can effectively reduce the size of the Cu_6_Sn_5_ precipitate phase and the thickness of the intermetallic layer, thereby improving the strength of the solder. However, In is prone to form a segregation effect with Ag in SAC105, forming new metal particles that hinder dislocation movement in the solder alloy, resulting in a significant reduction in elongation. Adding 2 wt.% Sb and 4.4 wt.% In to SAC105 simultaneously can increase the strength by 62.2% and still maintains high elongation. These better comprehensive mechanical properties can be attributed to the effect of Sb that forms a solid solution in the solder alloy, which can not only reduce the degree of In and Ag segregation but also promote the homogenization of the solder alloy structure.

With the addition of 0.3 wt.% Ni to SAC105-2Sb-4.4In solder, Ni can inhibit the growth of IMC Cu_3_Sn, the tensile strength is further increased to 54.0 MPa, which is almost 2.5 times the strength of SAC105 alloy, and the elongation is comparable to that of SAC105-2Sb-4.4In solder. As shown in Table 2, when Bi replaces Ni in SAC105-2Sb-4.4In-0.3Ni gradually, i.e., SAC105-2Sb-4.4In-0.2Ni-0.1Bi, SAC105-2Sb-4.4In-0.1Ni-0.2Bi, and SAC105-2Sb-4.4In-0.3Bi, the strength increases, and the elongation decreases gradually. SAC105-2Sb-4.4In-0.3Bi alloy owns the maximum strength of ~64.8 MPa, which is increased by ~185% than the SAC105, indicating that Bi has a positive effect on improving the strength of solder alloy. This is because Bi can effectively reduce the size of IMCs such as Ag_3_Sn and Cu_6_Sn_5_ [40], and 0.3 wt.% Bi can be completely dissolved in the β-Sn matrix [41], and the dispersion distribution plays a “pinning” role that can hinder the movement of dislocations, thereby improving the strength of the alloy. However, Bi addition will bring a decrease in elongation, SAC105-2Sb-4.4In-0.3Bi alloy owns the minimum elongation ~25.0%, which is decreased by ~89.2% with respect to SAC105. Moreover, Bi is easy to cause solder brittleness and excessive addition may lead to early failure of solders. From Figure 2, it is not difficult to find that, through the addition of Sb, In, Ni, and Bi and modulation of their contents, the UTS and elongation can be manipulated, and SAC105-2Sb-4.4In-0.3Ni, SAC105-2Sb-4.4In-0.2Ni-0.1Bi, SAC105-2Sb-4.4In-0.1Ni-0.2Bi and SAC105-2Sb-4.4In-0.3Bi solders have relatively good mechanical properties, especially SAC105-2Sb-4.4In-0.3Bi has the highest UTS with elongation higher than 20%, which meets the practical demands. Therefore, in the following, we will pay attention to the four solder alloys, which can be written as SAC105-2Sb-4.4In-(0.3 − *x*)Ni-*x*Bi (*x* = 0, 0.1, 0.2, 0.3).

The fracture morphologies of SAC105-2Sb-4.4In-(0.3 − *x*)Ni-*x*Bi solder alloys after the uniaxial tensile test were examined using SEM, as shown in Figure 3. Ductile dimples and pores are the main features that appear on the fracture surfaces of SAC samples, which is consistent with the good toughness of SAC solder alloys [42]. With the increase in Bi, cleavage surface and cleavage step appear on the fracture surfaces of the solder alloys with a few pits and cracks, indicating that the plasticity of the solder alloys decreases gradually. From Figure 3b–e, the range of the cleavage plane gradually increases, which is caused by the nucleation and expansion of cracks along the cleavage plane generated during the fracture of the alloy [43]. The decrease in dimple size is consistent with the increase in UTS and YS of the solder alloys and the decrease in elongation in the tensile test. Figure 3d shows the fracture morphology of SAC105-2Sb-4.4In-0.1Ni-0.2Bi solder alloy. A large number of cleavage steps, more holes, and smaller dimple size appear in the fracture, indicating increased brittleness, which is consistent with tensile properties in Figure 2, i.e., an increase in strength and a decrease in elongation. Figure 3e shows the fracture morphology of SAC105-2Sb-4.4In-0.3Bi solder alloy, which is composed of a cleavage plane, dimples, and several large holes, and the dimple size is further reduced, implying that the fracture mechanism is a combination of ductile fracture and brittle fracture.

### 3.2. Microstructure Characterization

XRD was conducted to identify the phase structures of SAC105 and SAC105-2Sb-4.4In-(0.3 − *x*)Ni-*x*Bi solders and the corresponding patterns are shown in Figure 4. It can be seen that all alloys consist of three main phases: the β-Sn phase, the Cu_6_Sn_5_ phase, and the Ag_3_Sn phase. No evident peaks were detected for Ag_3_Sn due to its low content. It is worth noting that when Bi is introduced into SAC105 alloy, the XRD peaks shift to the left, which may be attributed to the lattice distortion caused by Bi atoms being able to integrate into the Sn matrix. However, due to the low content of compounds containing Sb, In, Ni, and Bi, their existence cannot be detected from XRD patterns.

To further characterize the microstructure evolutions, SEM images are employed. Figure 5 shows the microstructures of SAC105 and SAC105-2Sb-4.4In-(0.3 − *x*)Ni-*x*Bi (*x* = 0, 0.1, 0.2, 0.3) solder alloys captured by SEM. SAC105 alloy shows a typical microstructure consisting of eutectic phases (dark regions) and primary β-Sn grains (bright regions), in which the eutectic phase consists of Ag_3_Sn and Cu_6_Sn_5_ IMCs, and the size of β-Sn grains and IMCs are large. When elements are added in SAC105, the microstructure is refined, i.e., the size of β-Sn grains and IMCs is obviously reduced. For example, 2 wt.% Sb is dissolved in the Sn matrix, which hinders the growth of IMCs and pushes them distributed more evenly. The 4.4 wt.% In would generate a new fine In_4_Ag_9_ IMCs, which can inhibit the dislocation movement and enhance the mechanical properties of solders. The 0.3 wt.% Ni would produce CuNi IMCs and has a certain effect on the strength improvement [44,45]. When Bi is substituted by Ni gradually, the microstructure of the alloys is refined obviously, and the IMCs (Ag_3_Sn and Cu_6_Sn_5_) are more evenly distributed in the Sn matrix. At the same time, due to the large size difference between Bi and Sn matrix, the solution strengthening effect caused by the addition of Bi will be stronger. Figure 6 shows the EDS analyses of SAC105 and SAC105-2Sb-4.4In-(0.3 − *x*)Ni-*x*Bi (*x* = 0, 0.1, 0.2, 0.3) alloys, with the additions of alloying elements, In_4_Ag_9_ and CuNi IMCs precipitated, it is worth noting that Bi dispersed in the Sn matrix of the alloy more evenly and did not form any IMCs. It can be expected that the strength increase caused by the addition of Bi will be more significant. The strength of SAC105-2Sb-4.4In-0.3Bi after the addition of 0.3 wt.% Bi is greater than that of SAC105-2Sb-4.4In-0.3Ni, which is consistent with the results of the tensile test (see Figure 2 and Table 2).

From the above results, we can see that the addition of Sb or Bi can cause lattice distortion in the crystal structure and increase the impending force to dislocation movement, thus improving the strength of the solder. The impending force can be expressed using the following equations [46]:(1)$$F_{m}=\mu br_{s}\left|\varepsilon \right|,$$
(2)$$\varepsilon =\left(r_{s}-r_{m}\right)/r_{m},$$
where *F*_m_ is the maximum force of the obstacle acting on the dislocation, *µ* is the modulus of rigidity, *b* is the Burgers vector, *ε* is the mismatch strain, *r*_s_ is the atomic radius of the solute, and *r*_m_ is the atomic radius in the matrix phase (Sn). As indicated in Equations (1) and (2), the force *F*_m_ is proportional to the mismatch strain *ε* that is associated with the difference in atomic radius between the solute and the matrix. Table 3 lists the atomic radius of the alloying elements involved in this study. It can be seen that the atomic size difference between Bi and Sn is larger than other alloying elements with Sn. Therefore, the solution strengthening effect caused by the addition of Bi is more pronounced. Moreover, the concentration of the solute element also influences the strengthening effect. It was reported that the solubility of Bi in Sn can be up to 3% [47]. Thus, adding 0.3 wt.% of Bi is within its solubility in Sn, Bi can completely distribute more homogeneous, and the solution strengthening effect will be more pronounced, which possibly explains the SAC105-2Sb-4.4In-0.3Bi solder alloy being with the highest UTS.

### 3.3. Thermal Property

The thermal property of the solder is of importance to the soldering quality, especially for SAC solders with relatively higher temperatures. Since high melting temperatures can increase the welding temperature, which has a negative impact on the quality of electronic products and devices. Therefore, it is necessary to maintain a low melting temperature and a small undercooling of the solder to improve the reliability of welding [48]. Figure 7 shows the DSC curves of SAC105, and SAC105-2Sb-4.4In-(0.3 − *x*)Ni-*x*Bi (*x* = 0, 0.1, 0.2, 0.3) solders, from which heating *T*_onset_, heating *T*_peak_ and cooling *T*_onset_ can be obtained. The difference between heating *T*_onset_ and cooling *T*_onset_ is the degree of undercooling required for nucleation in the solidification process, and the difference between heating *T*_onset_ and heating *T*_peak_ is defined as a pasty range [49,50]. In addition, some other small peaks appear on the DSC curve in the process of heating or cooling, which is stemmed from the addition of other elements to form IMCs, causing melt or precipitate faster in the process of heating or cooling. The specific data measured by DSC are summarized in Table 4.

As shown in Table 4, adding a certain amount of Sb, In, Ni, and Bi to SAC105 can reduce the undercooling of the solder alloy, thereby avoiding problems such as coarse intermetallic compound grains caused by high welding temperatures, and improving the reliability of the solder alloy [51]. From the heating *T*_onset_ values in Table 4, it can be observed that adding Sb, In and Ni can reduce the temperature of heating *T*_onset_ of SAC105, which decreases from 211.5 °C to 205.6 °C. After gradually replacing Ni with Bi, the melting point of the solder alloy continuously decreases. The melting point of SAC105-2Sb-4.4In-0.3Bi is the lowest ~202.0 °C, which is 4.5% lower than that of SAC105. During solidification, these Bi particles act as nucleation points, reducing the undercooling to 1.0 °C, which has a good melting performance. This indicates that the effect of Bi on reducing the melting point of solder is better than that of Ni, because Ni is easy to segregate with Cu and form CuNi IMC that is not easy to be melted, thus improving the melting point of the solder alloy. Bi does not react with the other elements, dispersing in the matrix of the alloy and playing a role in unifying the structure, thereby improving the melting properties of the solder alloy [52,53]. The pasty range represents the crystallization temperature of the alloys, and the probability of dispersive shrinkage is increased when the range expanded, which brings adverse effects on the properties. The synergistic action of alloying elements could increase the pasty range inevitably, however, the pasty range of SAC105-2Sb-4.4In-0.3Bi increased smaller as compared with other solder alloys, showing a relatively good melting performance.

### 3.4. Wettability

Wettability is related to the bonding quality of solders. Excellent wettability is a prerequisite for forming a reliable joint for industrial applications [54]. Wettability is determined by measuring the contact angle, *θ*, at the interface between the solder alloy and Cu substrate. A smaller contact angle means a better wettability of the solder alloy. Generally, the contact angle in the range of 0–30° is an excellent wetting, and the contact angle between 30° and 40° suggests a good wetting, while the contact angle larger than 70° indicates a low wettability [55]. Thus, to obtain good wettability, the contact angle *θ* should be between 0° and 40°. Figure 8 presents the contact angles of SAC105 and SAC105-2Sb-4.4In-(0.3 − *x*)Ni-*x*Bi (*x* = 0, 0.1, 0.2, 0.3) measured on Cu substrates, where the contact angles were averaged from three identical tests. It can be seen that the addition of Ni into SAC105 is prone to segregate with Cu, forming uneven Cu-Ni IMCs, resulting in stronger bonding forces within the alloy and thus, leading to an increase in the wetting angle to 40.8° as compared with the value of SAC105 (36.6°). When Ni is replaced by Bi, it can be seen that the contact angle of the alloy is correspondingly decreased although the contact angle ~37.9° for SAC105-2Sb-4.4In-0.2Ni-0.1Bi is a bit higher than that of SAC105 solder. As Bi content further increases, the contact angle becomes smaller and smaller. Since Bi is regarded as a stable element with the relative largest atomic mass, it can disperse and distribute on the substrate of the solder alloy showing a stable nature and hardly oxidizing. Therefore, adding Bi can prevent oxidation and reduce surface tension. Moreover, adding Bi can form weaker Sn-Bi or Sn-Sb bonds rather than stronger Sn-Sn bonds, which means that the bonding of internal forces is weakened, and the contact angle is reduced. Thus, the wetting angle of SAC105-2Sb-4.4In-0.3Bi is the smallest with a value of 34.7°, i.e., this solder alloy possesses the best wettability for the present study.

### 3.5. Creep Performance

To evaluate the creep behavior of SAC and SAC105-2Sb-4.4In-0.3Bi solder alloys, tensile creep was conducted at different temperatures under constant tensile stress (12 MPa) and at a constant temperature (25 °C) with different applied stresses, and the corresponding creep curves are shown in Figure 9. The shape of creep curves shows well-defined three distinct creep stages, i.e., primary creep, secondary steady-state creep, and tertiary creep. Apparently, the creep behavior exhibits two features: (i) regardless of the composition of the solders, the higher the creep temperature or stress, the higher the creep rate, and the shorter the creep lifetime; (ii) with respect to SAC105, the creep rate and the creep lifetime of SAC105-2Sb-4.4In-0.3Bi solder at a given temperature or stress are slower and longer, respectively.

To better compare the creep resistance of SAC105 and SAC105-2Sb-4.4In-0.3Bi, their steady-state creep strain rates at different temperatures and different stresses, $\dot{\varepsilon }$, were computed by linear fitting, as summarized in Table 5 and Table 6, respectively. Remarkably, the steady-state creep strain rate of SAC105-2Sb-4.4In-0.3Bi at a specific temperature is significantly decreased, which is one order magnitude lower than that of SAC105. Owing to the low creep strain rate, the creep lifetime of SAC105-2Sb-4.4In-0.3Bi is much longer with approximately one order magnitude as compared to SAC105 (see Table 5). For the creep test at room temperature with different applied stresses, SAC105-2Sb-4.4In-0.3Bi solder also has a lower steady-state creep strain rate and longer creep lifetime, which can be up to two order magnitude for low applied stresses such as 12 MPa and 14 MPa (see Table 6). These data strongly indicate that the addition of Sb, In, and Bi elements are beneficial to improve the creep strength of SAC105 solder and could enhance its creep resistance.

Due to the homologous temperature of SAC105-based solders at room temperature exceeding 0.5, creep should be the dominant deformation-controlling mechanism. Since the secondary steady-state stage takes up the majority of the creep time to the final failure of the solder, the steady-state creep region is often chosen to represent the whole process in the constitutive models. In this stage, the strain rate, $\dot{\varepsilon }$, one of the most significant parameters of creep resistance in engineering assessments, follows the Norton power law [56]:(3)$$\dot{\varepsilon }=A\sigma^{n}\exp\left(\frac{-Q}{RT}\right),$$
where *A* is a constant independent of temperature, *σ* is the applied stress, *n* is the stress exponent constant related to strain rate, *Q* is the activation energy, *R* is the universal gas constant, and *T* is the absolute temperature. Taking the natural logarithmic transformation on both sides of Equation (3), then we have:(4)$$ln\dot{\varepsilon }=lnA+nln\sigma -\frac{Q}{RT}.$$

Therefore, the stress exponent (*n*) can be obtained from the slope of the *ln* ($\dot{\varepsilon }$) against *ln* (*σ*) plot and the creep activation energy (*Q*) can be calculated by linear regression of experimental data, as displayed in Figure 10. The results show that, with the additions of Sb, In, and Bi, the *n* value is increased to 11.1, which is higher than that of SAC105 (*n* = 7.8), suggesting a better creep resistance for SAC105-2Sb-4.4In-0.3Bi solder. Different creep mechanisms, such as grain boundary sliding, lattice diffusion, and dislocation climbing and gliding, have been proposed. If the *n* value is less than 3, grain boundary sliding dominates the creep deformation. If the *n* value is greater than 3, dislocation climbing and gliding is the major acting mechanism [57]. Therefore, in our case, the creep process is predominantly by the dislocation climbing and gliding mechanism. The addition of Sb, In, and Bi could improve the creep resistance by two strengthening mechanisms: (1) The microstructure and precipitates such as Ag_3_Sn, Cu_6_Sn_5_, and In_4_Ag_9_ are refined and distributed more uniformly, which can restrict the glide of dislocations; (2) The solutes such as Sb and Bi are dissolved in the matrix, which can cause lattice distortion and impede the dislocation movement and enhance the creep resistance. At the same time, the activation energy is also increased from 95.4 kJ/mol to 111.8 kJ/mol, which is approximately enhanced by ~17.2%, indicating a better creep resistance for SAC105-2Sb-4.4In-0.3Bi solder. It is interesting to note that the values of creep activation energy obtained in the present study are very close to those for the lattice diffusion of pure Sn that reported between 94–118 kJ/mol [58,59,60], which implies that lattice diffusion is the main creep mechanism for the currently studied solder alloys.

## 4. Conclusions

In this study, the alloying effects of Sb, In, Ni, and Bi on the microstructure, mechanical properties, melting point, wettability, and creep resistance of SAC105-2Sb-4.4In-(0.3 − *x*)Ni-*x*Bi (*x* = 0, 0.1, 0.2, 0.3) solder alloys were investigated. Based on the results, we can draw the following conclusions:When Sb, In, Ni, and Bi were added to SAC105, the microstructure of solder alloys was refined. Sb and Bi can dissolve into the Sn matrix without forming any precipitations, which has a solid solution strengthening effect. The eutectic zones composed of Ag_3_Sn and Cu_6_Sn_5_ were refined and distributed more evenly in the SAC105-2Sb-4.4In-0.3Bi as compared with other compositions, implying Bi shows a better refinement effect.By adding 2 wt.% Sb and 4.4 wt.% In, the tensile strength of SAC105 was increased from 27.9 MPa to 36.8 MPa. When 0.3 wt.% Ni was further added to SAC105, the tensile strength can be further increased to 54.4 MPa, which indicates that Ni has a significant effect on the improvement of the solder alloy. When Bi substitutes Ni gradually, the tensile strength was further improved. It was found that SAC105-2Sb-4.4In-0.3Bi alloy owns the highest strength ~64.8 MPa which is approximately 185% higher than that of SAC105. At the same time, the elongation of SAC105-2Sb-4.4In-0.3Bi remains at an acceptable level (>20%), which meets the actual production requirements.The alloying elements could improve the melting properties by considerably reducing the undercooling of solder alloys. It was found that SAC105-2Sb-4.4In-0.3Bi alloy has the smallest undercooling ~1.0 °C which is much smaller than that of SAC105. Although its pasty range was enlarged inevitably, the pasty range of SAC105-2Sb-4.4In-0.3Bi increased less as compared with other solder alloys, showing a relatively good melting performance.The wettability test conducted on the Cu substrate shows that the addition of Sb, In, and Bi can improve the wettability of the SAC105 alloy. The wetting angle of SAC105 is 36.6°, while the wetting angle of SAC105-2Sb-4.4In-0.3Bi decreases to 34.7°. Moreover, the addition of Bi has a stronger effect on improving wettability than Ni.Adding Sb, In, and Bi to SAC105 can improve the creep resistance significantly. Through DMA creep testing, it was found that SAC105-2Sb-4.4In-0.3Bi has a better creep resistance as compared with SAC105. This can be attributed to the synergistic alloying effect of Sb, In, and Bi, which improves the creep resistance through solid solution strengthening and precipitation strengthening mechanisms.

## Figures and Tables

**Figure 1 materials-16-04059-f001:**
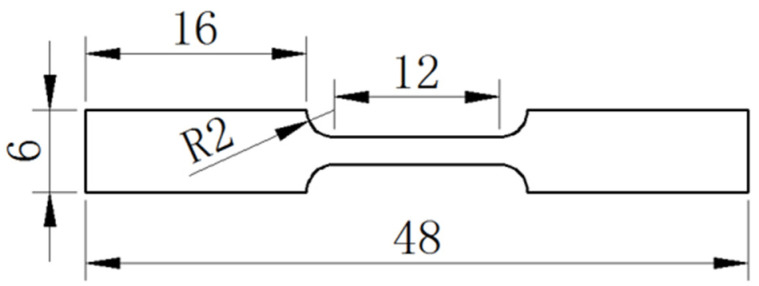
Schematic of the tensile test specimen (unit: mm).

**Figure 2 materials-16-04059-f002:**
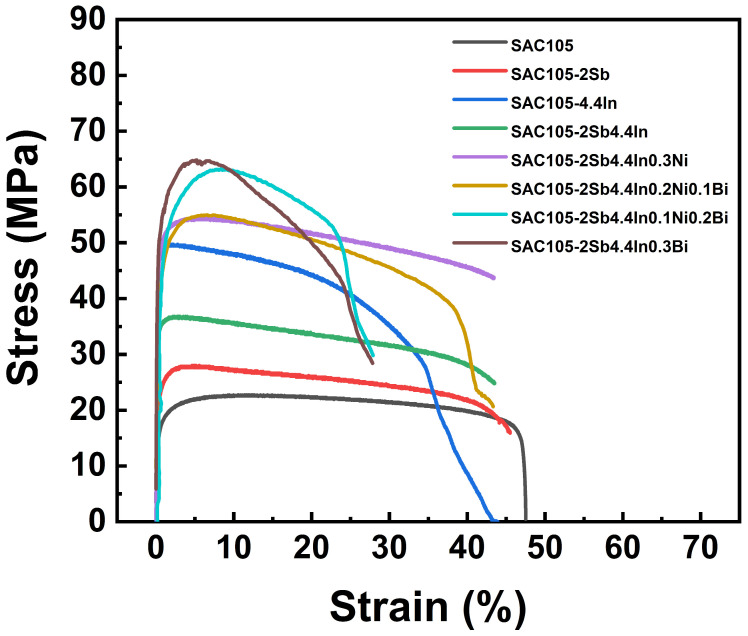
Stress–strain curves of SAC105 solders with different element additions.

**Figure 3 materials-16-04059-f003:**
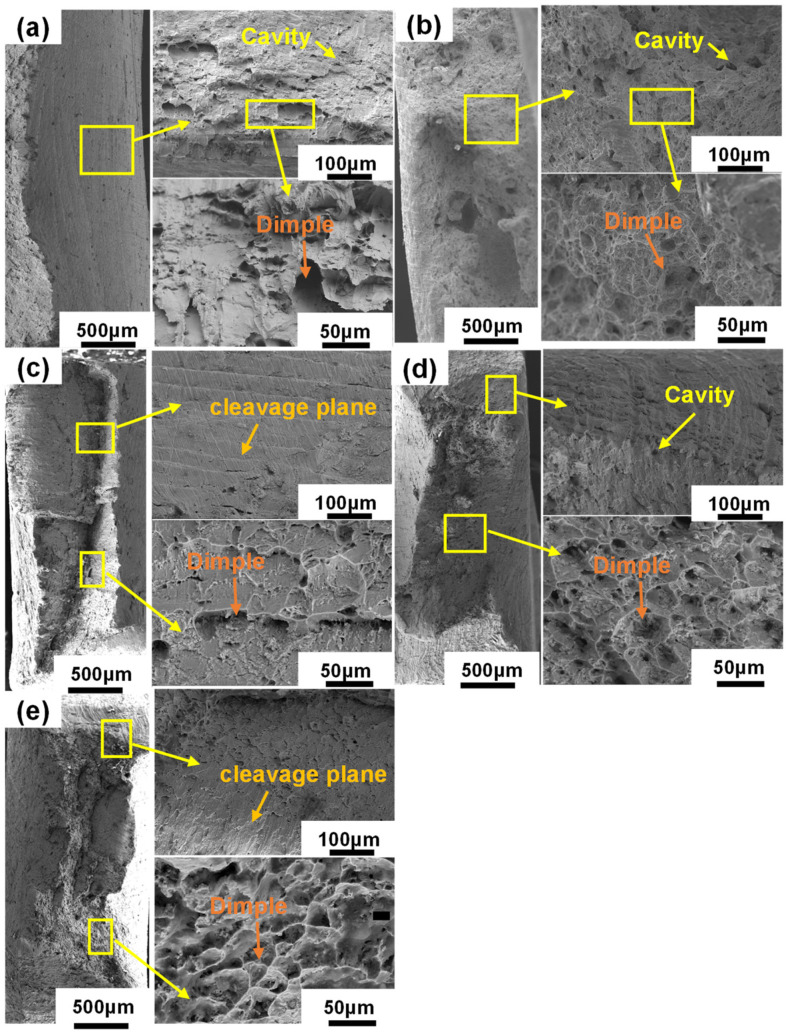
Fracture morphologies of alloys with different compositions. (**a**) SAC105, (**b**) SAC105-2Sb-4.4In-0.3Ni, (**c**) SAC105-2Sb-4.4In-0.2Ni-0.1Bi, (**d**) SAC105-2Sb-4.4In-0.1Ni-0.2Bi, and (**e**) SAC105-2Sb-4.4In-0.3Bi.

**Figure 4 materials-16-04059-f004:**
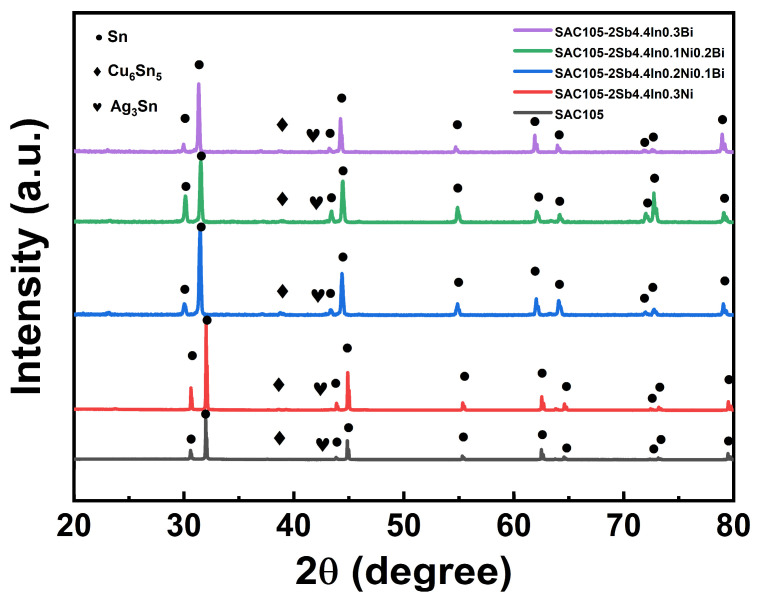
XRD patterns of SAC105 and SAC105-2Sb-4.4In-(0.3 − *x*)Ni-*x*Bi (*x* = 0, 0.1, 0.2, 0.3).

**Figure 5 materials-16-04059-f005:**
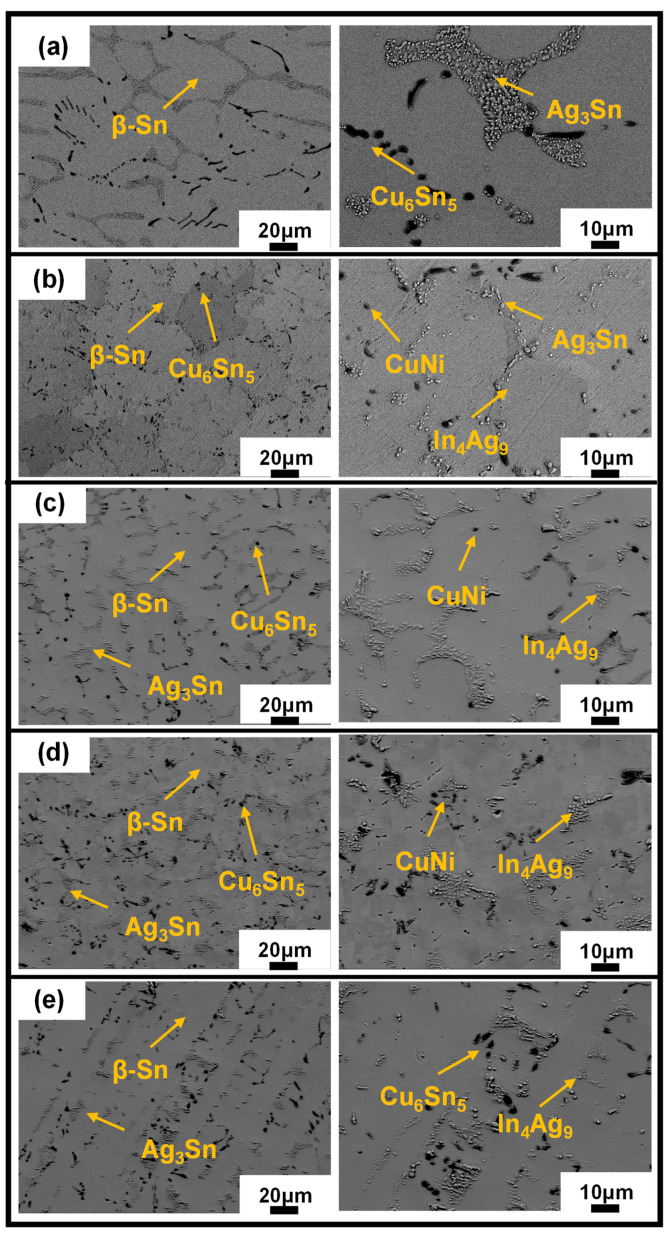
Microstructure of the solder alloys with different compositions. (**a**) SAC105, (**b**) SAC105-2Sb-4.4In-0.3Ni, (**c**) SAC105-2Sb-4.4In-0.2Ni-0.1Bi, (**d**) SAC105-2Sb-4.4In-0.1Ni-0.2Bi, and (**e**) SAC105-2Sb-4.4In-0.3Bi.

**Figure 6 materials-16-04059-f006:**
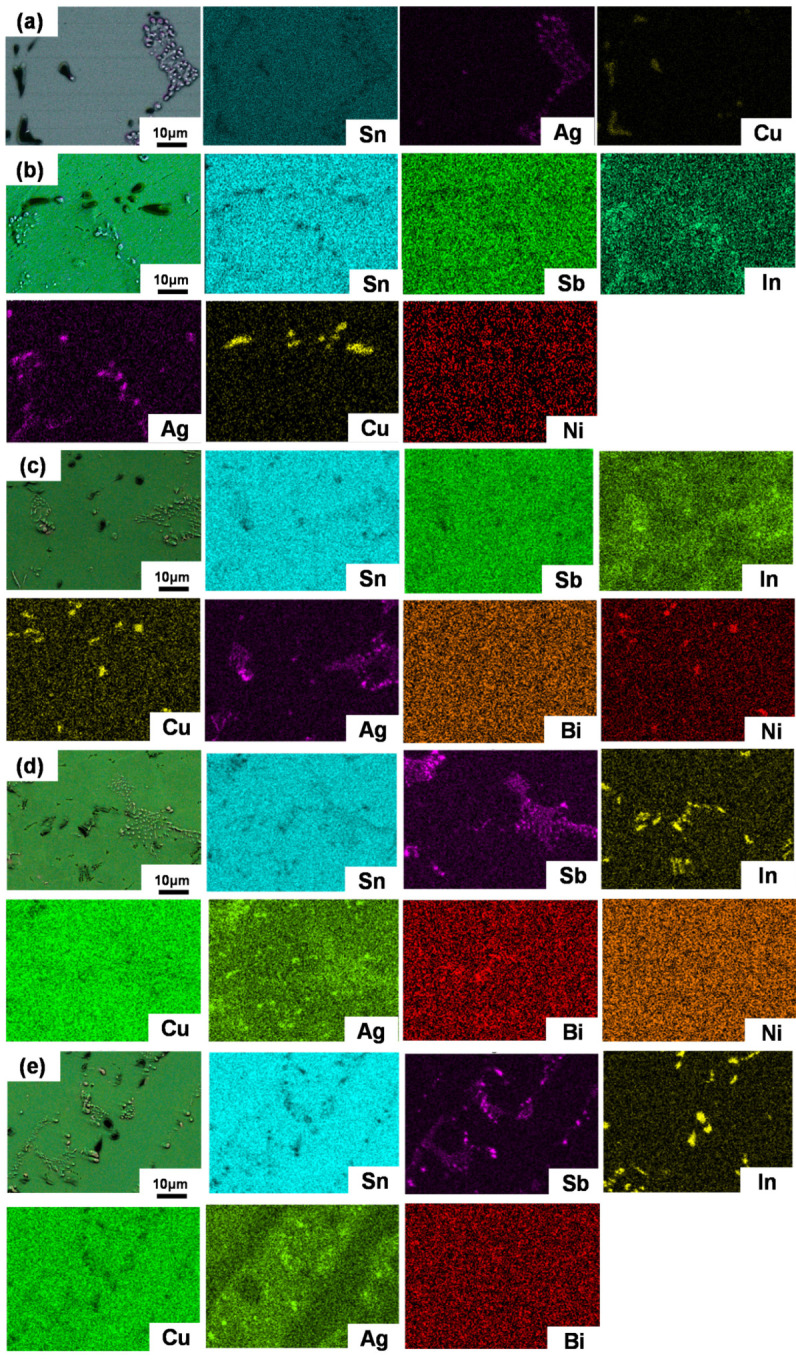
EDS elemental mapping of the solder alloys. (**a**) SAC105, (**b**) SAC105-2Sb-4.4In-0.3Ni, (**c**) SAC105-2Sb-4.4In-0.2Ni-0.1Bi, (**d**) SAC105-2Sb-4.4In-0.1Ni-0.2Bi, and (**e**) SAC105-2Sb-4.4In-0.3Bi.

**Figure 7 materials-16-04059-f007:**
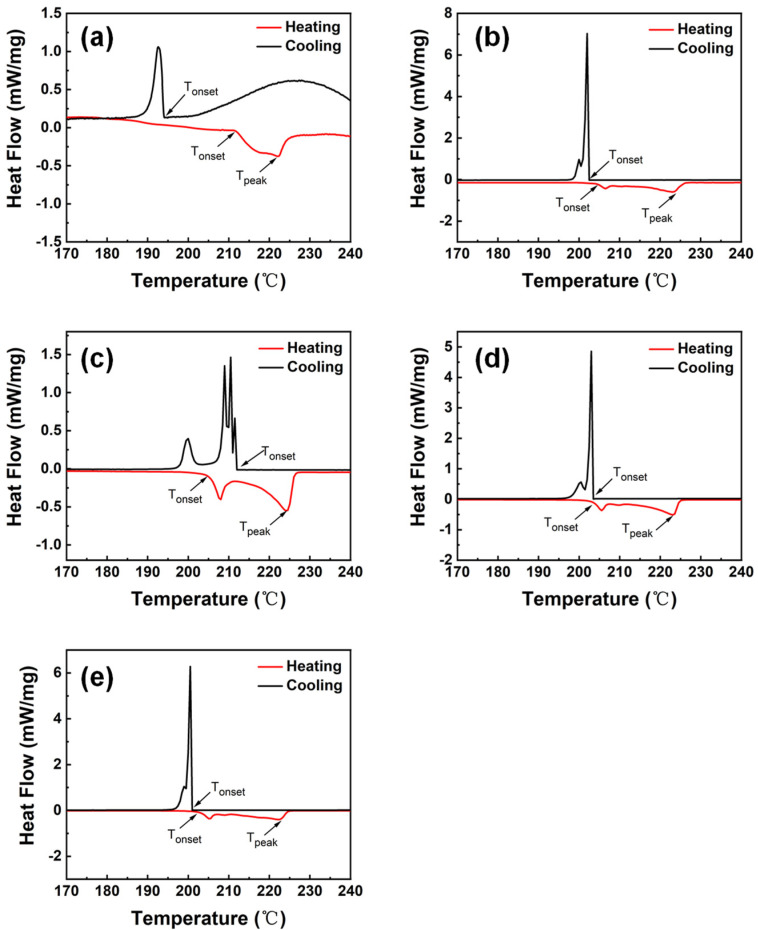
DSC curves of solders with different components. (**a**) SAC105, (**b**) SAC105-2Sb-4.4In-0.3Ni, (**c**) SAC105-2Sb-4.4In-0.2Ni-0.1Bi, (**d**) SAC105-2Sb-4.4In-0.1Ni-0.2Bi, and (**e**) SAC105-2Sb-4.4In-0.3Bi.

**Figure 8 materials-16-04059-f008:**
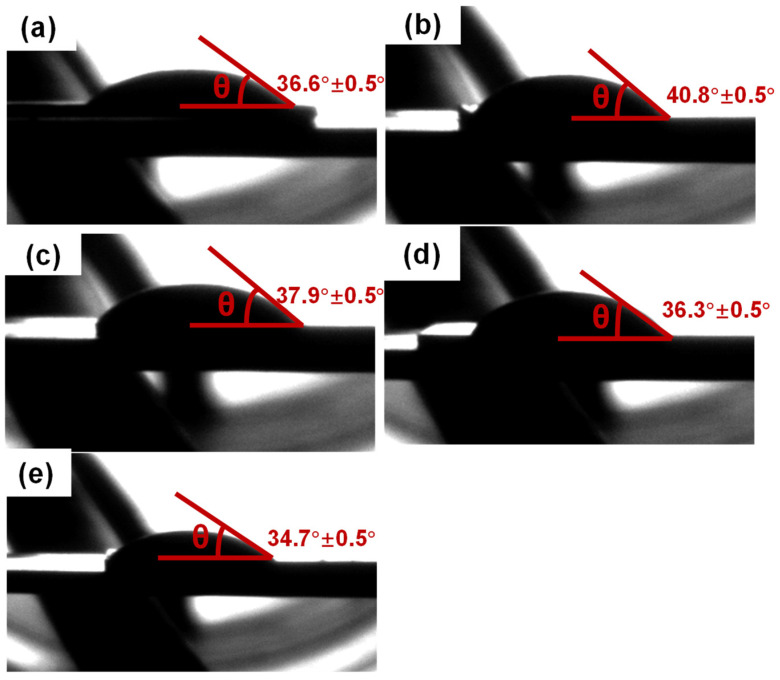
Measurement of contact angles of the solder alloys with different components. (**a**) SAC105, (**b**) SAC105-2Sb-4.4In-0.3Ni, (**c**) SAC105-2Sb-4.4In-0.2Ni-0.1Bi, (**d**) SAC105-2Sb-4.4In-0.1Ni-0.2Bi, and (**e**) SAC105-2Sb-4.4In-0.3Bi.

**Figure 9 materials-16-04059-f009:**
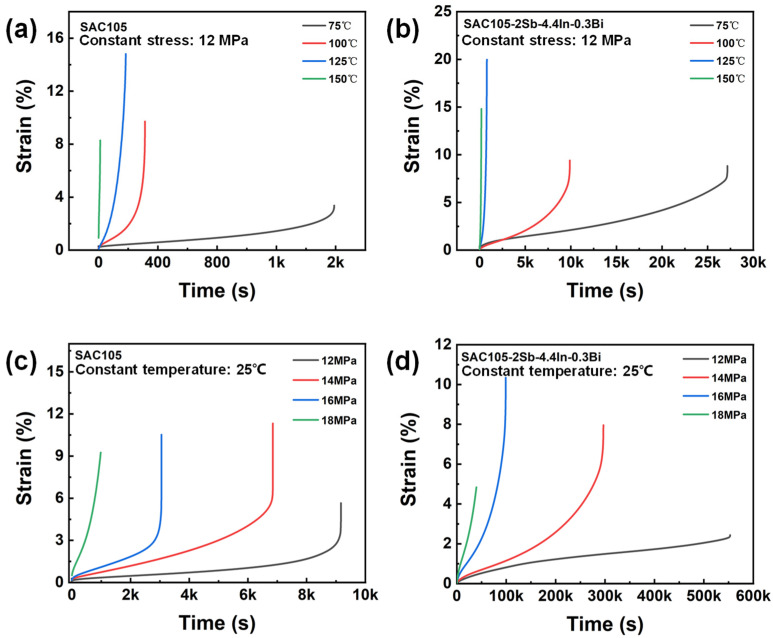
Comparison of creep curves of SAC105 and SAC105-2Sb-4.4In-0.3Bi at different temperatures under a constant tensile stress of 12 MPa (**a**,**b**) and different applied stresses at a constant temperature of 25 °C (**c**,**d**).

**Figure 10 materials-16-04059-f010:**
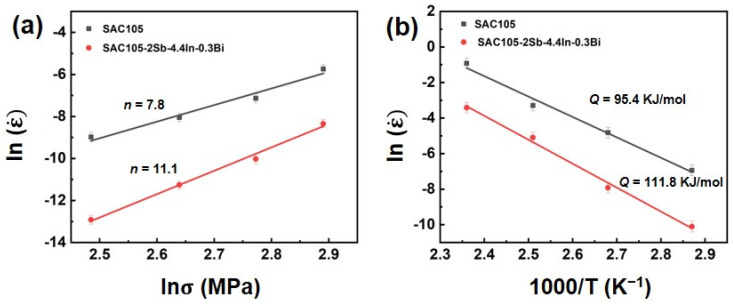
(**a**) Relationship between ln (σ) and ln ($\dot{\varepsilon }$) at room temperature, and (**b**) temperature dependence of steady-state creep rate of SAC105 and SAC105-2Sb-4.4In-0.3Bi solder alloys.

**Table 1 materials-16-04059-t001:** Compositions of SAC105-based solder alloys (wt.%).

Alloy (wt.%)	Ag	Cu	Sb	In	Ni	Bi	Sn
SAC105	1.0	0.5	2.0	4.4			Bal.
SAC105-2Sb	1.0	0.5	2.0				Bal.
SAC105-4.4In	1.0	0.5		4.4			Bal.
SAC105-2Sb-4.4In	1.0	0.5	2.0	4.4			Bal.
SAC105-2Sb-4.4In-0.3Ni	1.0	0.5	2.0	4.4	0.3		Bal.
SAC105-2Sb-4.4In-0.2Ni-0.1Bi	1.0	0.5	2.0	4.4	0.2	0.1	Bal.
SAC105-2Sb-4.4In-0.1Ni-0.2Bi	1.0	0.5	2.0	4.4	0.1	0.2	Bal.
SAC105-2Sb-4.4In-0.3Bi	1.0	0.5	2.0	4.4		0.3	Bal.

**Table 2 materials-16-04059-t002:** Mechanical properties of the solders with different components.

Component (wt.%)	UTS (MPa)	YS (MPa)	Elongation (%)
SAC105	22.7 ± 2.6	13.8 ± 1.8	47.1 ± 2.1
SAC105-2Sb	27.9 ± 4.9	20.1 ± 3.7	45.4 ± 4.4
SAC105-4.4In	49.6 ± 3.2	45.2 ± 2.9	36.4 ± 3.8
SAC105-2Sb-4.4In	36.8 ± 5.5	33.6 ± 4.3	43.4 ± 6.2
SAC105-2Sb-4.4In-0.3Ni	54.0 ± 3.4	37.9 ± 3.6	43.4 ± 4.5
SAC105-2Sb-4.4In-0.2Ni-0.1Bi	54.5 ± 4.9	39.1 ± 3.9	40.8 ± 3.7
SAC105-2Sb-4.4In-0.1Ni-0.2Bi	63.1 ± 2.8	41.4 ± 2.3	26.1 ± 2.9
SAC105-2Sb-4.4In-0.3Bi	64.8 ± 3.1	50.2 ± 2.6	25.0 ± 2.4

**Table 3 materials-16-04059-t003:** Atomic radius of different alloying elements.

Elements	Sn	Sb	In	Ni	Bi
Atomic radius (nm)	0.158	0.160	0.166	0.124	0.170

**Table 4 materials-16-04059-t004:** Melting properties of SAC105-based solder alloys.

Alloys	Heating *T*_onset_ (°C)	Heating *T*_peak_ (°C)	Cooling *T*_onset_ (°C)	Undercooling (°C)	Pasty Range (°C)
SAC105	211.5 ± 3.1	221.9 ± 1.8	194.0 ± 3.2	17.5 ± 1.9	10.4 ± 2.1
SAC105-2Sb-4.4In-0.3Ni	205.6 ± 2.5	224.7 ± 2.7	202.5 ± 2.4	3.1 ± 2.4	19.1 ± 3.7
SAC105-2Sb-4.4In-0.2Ni-0.1Bi	205.1 ± 2.9	224.5 ± 2.1	212.0 ± 3.7	6.9 ± 1.7	19.5 ± 4.5
SAC105-2Sb-4.4In-0.1Ni-0.2Bi	204.6 ± 3.4	223.7 ± 3.7	202.5 ± 2.1	2.1 ± 1.3	19.2 ± 3.2
SAC105-2Sb-4.4In-0.3Bi	202.0 ± 2.3	220.9 ± 2.6	201.0 ± 3.2	1.0 ± 0.8	18.9 ± 2.9

**Table 5 materials-16-04059-t005:** Creep rate and creep lifetime at different temperatures under a constant applied stress of 12 MPa.

Temperature (°C)	SAC105		SAC105-2Sb-4.4In-0.3Bi	
	Creep Rate (s^−1^)	Creep Life (s)	Creep Rate (s^−1^)	Creep Life (s)
75	9.6 × 10^−4^ ± 3.1 × 10^−8^	1.6 × 10^3^ ± 4.1 × 10^1^	4.1 × 10^−5^ ± 9.2 × 10^−8^	2.7 × 10^4^ ± 8.9 × 10^2^
100	8.1 × 10^−3^ ± 5.7 × 10^−7^	3.1 × 10^2^ ± 3.7 × 10^1^	3.6 × 10^−4^ ± 7.9 × 10^−7^	9.9 × 10^3^ ± 6.3 × 10^1^
125	3.7 × 10^−2^ ± 6.4 × 10^−5^	1.8 × 10^2^ ± 9.1 × 10^0^	6.1 × 10^−3^ ± 4.2 × 10^−5^	7.9 × 10^2^ ± 7.7 × 10^0^
150	3.9 × 10^−1^ ± 4.9 × 10^−3^	1.1 × 10^1^ ± 0.5 × 10^0^	3.3 × 10^−2^ ± 3.7 × 10^−4^	1.8 × 10^2^ ± 5.6 × 10^0^

**Table 6 materials-16-04059-t006:** Creep rate and creep lifetime under different stresses at a constant temperature of 25 °C.

Stress (MPa)	SAC105		SAC105-2Sb-4.4In-0.3Bi	
	Creep Rate (s^−1^)	Creep Life (s)	Creep Rate (s^−1^)	Creep Life (s)
12	1.3 × 10^−4^ ± 4.2 × 10^−8^	9.2 × 10^3^ ± 7.3 × 10^1^	2.5 × 10^−6^ ± 5.1 × 10^−10^	5.5 × 10^5^ ± 4.7 × 10^2^
14	3.2 × 10^−4^ ± 4.9 × 10^−7^	6.9 × 10^3^ ± 5.6 × 10^1^	1.3 × 10^−5^ ± 2.1 × 10^−8^	2.9 × 10^5^ ± 2.9 × 10^2^
16	7.9 × 10^−4^ ± 4.1 × 10^−7^	3.1 × 10^3^ ± 2.9 × 10^1^	4.5 × 10^−5^ ± 6.7 × 10^−8^	9.9 × 10^4^ ± 2.3 × 10^2^
18	3.2 × 10^−3^ ± 3.4 × 10^−5^	9.8 × 10^2^ ± 8.7 × 10^0^	2.4 × 10^−4^ ± 5.2 × 10^−8^	3.9 × 10^4^ ± 1.5 × 10^1^

## Data Availability

Data are contained within the article.

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
