# Peer review of "Investigation of Microstructure and Mechanical Properties of SAC105 Solders with Sb, In, Ni, and Bi Additions"

_materials, 2023, doi:10.3390/ma16114059_

Round 1
Reviewer 1 Report
Reviewer comments
This work looks at the role of alloying elements on the microstructure, mechanical properties, melting point, wettability, and creep resistance of SAC-105. The manuscript is interesting and would be interesting to the readers of the journal. I recommend publishing after addressing these minor revisions.
Abstract:
Line 1: SAC not defined in the first use.
Line 13: IMCs not defined in the first use. Please use intermetallic compounds
Introduction:
Line 33: it is “considered as a substitute.”
Line 42: it is “decrease in Ag”
Line 54: it is “to form solid solution”
Line 63: it is “could diffuse”
Line 68: it is “are increased”
Line 74: strike out up to now and start the sentence with Even though
Line 78: change “impact of additions of Sb, In, Ni, and Bi” to “impact of Sb, In, Ni, and Bi additions on”
Experimental Methods:
Line 99: Not clear what does the author mean by 5 parallel specimens? Is it 5 similar specimens?
Line 109: hypen between energy-dispersive
Line 119: in some places spaces were used between values and oC units and in some cases not. I guess there shouldn’t be space between values and degree Celsius. Please make changes throughout the manuscript.
Line 122: replace with a weight with” sample weighing 15-30 mg”
Line 134: From which doesn’t seem correct, please consider changing to From this,
Line 160: 2.5 times of the strength; remove “of” from this sentence.
Line 185: it should be “decreases gradually”
Line 193: it should be “i.e, an increase of strength”
Figure 4: In the legends should be number should be subscript.
Figure 4: There is no evident peaks for Ag3Sn, can you please explain this.
Figure 6: Why was EDS mapping of other alloys not shown?
Line260: With the relative higher temperature should be “with relatively higher temperature”
Figure 7 (a), Why is the cooling curve for figure 7(a) showing an increase in the heat flow at higher temperatures? Whereas this is absent in all the other cases.
Line 300: “application” should be “applications”.
Line 307,308; not clear what the author means by “on the basis of SAC105”
Line 311: hardly oxidized to hardly oxidizing.
Line 333: remove both is from the sentence “higher is the creep rate,” same with other “is” in the whole sentence.
Have included in the comments to authors
Author Response
Dear reviewer,
We highly appreciate the time and efforts you have dedicated to providing valuable feedback. We are sorry that some presentations are still not quite clear and confused in the last manuscript. We have carefully addressed the issues raised by the reviewers, point by point. All changes in the revised manuscript are highlighted by red color. We believe your comments and suggestions have significantly improved the quality of our manuscript and made it publishable.
Comment 1:
Line 1: SAC not defined in the first use.
Line 13: IMCs not defined in the first use. Please use intermetallic compounds.
Response: Many thanks for the reviewer pointing out our missing part. We have revised SAC to Sn-Ag-Cu (SAC) and IMCs to intermetallic compounds, respectively. Please see the revised abstract highlighted by red color.
Comment 2:
Line 33: it is “considered as a substitute.”
Line 42: it is “decrease in Ag”
Line 54: it is “to form solid solution”
Line 63: it is “could diffuse”
Line 68: it is “are increased”
Line 74: strike out up to now and start the sentence with Even though
Line 78: change “impact of additions of Sb, In, Ni, and Bi” to “impact of Sb, In, Ni, and Bi additions on”
Response: Thank you very much for the comments. We have revised the typographical and grammatical errors which are highlighted by red color. Please see the revised manuscript.
Comment 3:
Line 99: Not clear what does the author mean by 5 parallel specimens? Is it 5 similar specimens?
Response: We are sorry for our unclear statement. “5 parallel specimens” mean compression test was conducted 5 times for each alloy composition to ensure repeatability. We have changed the wording, please see Line 99.
Comment 4:
Line 109: hypen between energy-dispersive
Line 119: in some places spaces were used between values and ℃ units and in some cases not. I guess there shouldn’t be space between values and degree Celsius. Please make changes throughout the manuscript.
Line 122: replace with a weight with” sample weighing 15-30 mg”
Line 134: From which doesn’t seem correct, please consider changing to From this,
Line 160: 2.5 times of the strength; remove “of” from this sentence.
Line 185: it should be “decreases gradually”
Line 193: it should be “i.e, an increase of strength”
Response: Thank you for your suggestions. We have made all the corrections based on your comments.
Comment 5:
Figure 4: In the legends should be number should be subscript.
Figure 4: There is no evident peaks for Ag3Sn, can you please explain this.
Response: Thank you for your comments. We have modified the legends in Figure 3. Low content of Ag will lead to the low content of Ag3Sn, correspondingly causing their insignificant peaks in XRD patterns. The peak of Ag3Sn will be evident in SAC205 for relative high Ag content [Materials Design, 2013, 47:607-614].
Comment 6:
Figure 6: Why was EDS mapping of other alloys not shown?
Response: Thanks for the reviewer pointing the missing part. We have added the EDS mappings of other solder alloys. Please see Lines 226-229.
Comment 7:
Figure 7 (a), Why is the cooling curve for figure 7(a) showing an increase in the heat flow at higher temperatures? Whereas this is absent in all the other cases.
Response: Thank you for your comment. The reason may be that during the cooling process (Figure 7(a)), the microstructure of Cu6Sn5 and Ag3Sn IMCs in the SAC105 are coarse and difficult to cool down, so the heat flow increases. However, after alloying (the other cases in Figure 7), the microstructure of Cu6Sn5 and Ag3Sn IMCs in the solder are refined and easy to cool down, so the phenomenon of an increase in heat flow does not occur.
Comment 8:
Line 260: With the relative higher temperature should be “with relatively higher temperature”
Line 300: “application” should be “applications”.
Line 307,308; not clear what the author means by “on the basis of SAC105”
Line 311: hardly oxidized to hardly oxidizing.
Line 333: remove both is from the sentence “higher is the creep rate,” same with other “is” in the whole sentence.
Response: Thank you very much for your comments. We have changed the unclear wording “on the basis of SAC105” to “into SAC105”. For the typographical and grammatical errors, we have revised and highlighted by red color.
We would like to thank you again for your valuable and constructive comments.
Sincerely yours,
Xilei Bian

Reviewer 2 Report
This is a well thought out study of tin solder development, and the authors present a large set of experimental results. However, the manuscript has two major weaknesses:
1. All experimental data are single specimen measurements with no discussion about experimental errors and inaccuracies. Because there are no repetitions, there is no statistical strength to the study and the reader is left wondering how accurate the presented values are.
2. Referencing is poorly done throughout the manuscript. In multiple occasions when a method, equation or findings are referenced, the authors seem to point to some secondary source instead of the original source. An example would be at the beginning of the introduction: talking about a publication of the US EPA regarding harmful chemicals, the reference points to Anderson et al about the effects of alloying. In the same paragraph, the authors list SAC305, SAC387 and SAC396 materials with their properties, but at the end of the sentence, references 7-9 are pointing to works on different alloys. I would like to ask the authors to please carefully review their references throughout the paper. In the end, references 55-60 are not even listed.
Additional notes:
Figure 2 and Table 2: The dogbone tensile testing by UTM study is a prime example when single sample testing is insufficient. Multiple samples of the same type of specimen are necessary to elicit the tensile behavior. Were the measurements run at room temperature? How was Yield Strength calculated? The YS numbers in Table 2 do not seem to match the plots in Figure 2.
Figure 3 and Figure 5: The SEM images are way too tiny for the reader to observe any features that are discussed in the manuscript.
Figure 7 and Table 4: Again, the presented numbers are from a single measurement instead of being averages. If there was no way to measure multiple samples of the same type of specimen, then the authors must disclose the estimated error on the presented numbers.
Figure 8: What is the accuracy of the wetting angle measurements? Do the authors believe that they have tenth of a degree accuracy? Does the difference in wetting angle measurement results can really be a claimable conclusion?
Figure 9 and Table 5: Besides the same criticism about the lack of statistics, What is the stress applied in Figure 9 a) and b)? What is the temperature applied in Figure 9 c) and d)? How is creep life calculated? Especially at high temperatures vs lower temperatures.
Conclusions:
On Line 405 "highest strength ~64.8 MPa that is approximately 132% higher than that of SAC105". This is strange. 64.8/22.7 = 285% in my calculation.
The English language of the manuscript is good!
Author Response
Dear reviewer,
We highly appreciate the time and efforts you have dedicated to providing valuable feedback. We are sorry that some presentations are still not quite clear and confused in the last manuscript. We have carefully addressed the issues raised by the reviewers, point by point. All changes in the revised manuscript are highlighted by red color. We believe your comments and suggestions have significantly improved the quality of our manuscript and made it publishable.
Comment 1:
All experimental data are single specimen measurements with no discussion about experimental errors and inaccuracies. Because there are no repetitions, there is no statistical strength to the study and the reader is left wondering how accurate the presented values are.
Response: We are very sorry for our mistake and unclear statement. Actually, in our study, the test for each alloy composition was carried out for 3~5 times, and similar statement can be found in Line 99 in our original manuscript. For the experimental errors and inaccuracies, we have to admit that it is indeed our mistake. We have added the experimental errors in the revised manuscript. The related figures of repeatability tests are shown at the back attachment.
Comment 2:
Referencing is poorly done throughout the manuscript. In multiple occasions when a method, equation or findings are referenced, the authors seem to point to some secondary source instead of the original source. An example would be at the beginning of the introduction: talking about a publication of the US EPA regarding harmful chemicals, the reference points to Anderson et al about the effects of alloying. In the same paragraph, the authors list SAC305, SAC387 and SAC396 materials with their properties, but at the end of the sentence, references 7-9 are pointing to works on different alloys. I would like to ask the authors to please carefully review their references throughout the paper. In the end, references 55-60 are not even listed.
Response: We are so sorry for our mistake and thank you very much the reviewer’s comment. We have added the original reference sources and filled the missing references. Please see the revised manuscript.
Comment 3:
Figure 2 and Table 2: The dog-bone tensile testing by UTM study is a prime example when single sample testing is insufficient. Multiple samples of the same type of specimen are necessary to elicit the tensile behavior. Were the measurements run at room temperature? How was Yield Strength calculated? The YS numbers in Table 2 do not seem to match the plots in Figure 2.
Response: (1) We have revised the content of the manuscript. In our study, the dog-bone tensile tests were conducted at least 3 times of each sample (each sample was made 3 repetitions). The related charts of repeatability tests are shown in the attachment.
(2) The tensile tests were carried out at room temperature, as we have already stated clearly in Line 95 in our original manuscript.
(3) Yield strength (YS) were calculated by the stress value corresponding to 0.2% plastic strain. The reason why “The YS numbers in Table 2 do not seem to match the plots in Figure 2” is that Figure 2 shows the typical curves of four repetitions, but in the Table 2, we listed the average YS value of four repetitions.
Comment 4:
Figure 3 and Figure 5: The SEM images are way too tiny for the reader to observe any features that are discussed in the manuscript.
Response: Thank you very much for your valuable comment. We have revised the SEM images in the revised manuscript that could be clearly for the reader to observe any features. Please see the SEM images in the revised Figure 3 and Figure 5.
Comment 5:
Figure 7 and Table 4: Again, the presented numbers are from a single measurement instead of being averages. If there was no way to measure multiple samples of the same type of specimen, then the authors must disclose the estimated error on the presented numbers.
Response: Thank you for your comment. In our study, the melting properties were tested by differential scanning calorimetry (DSC), which owns high precision (0.1 μW), so each sample was tested for one time. We have disclosed the estimated errors on the presented values in Table 4.
Comment 6:
Figure 8: What is the accuracy of the wetting angle measurements? Do the authors believe that they have tenth of a degree accuracy? Does the difference in wetting angle measurement results can really be a claimable conclusion?
Response: Thank you for your comment. The wettability tests were performed three times for each alloy composition, then we chose a typical specimen from each composition and took photos by the wettability angle tester. Angles were measured by the Image J software which owns high precision, and we reserved a decimal fraction of the data. The wetting angle mainly reflects the wettability of solders, then, from Figure 8, we can compare the effect of microalloying on wettability.
Comment 7:
Figure 9 and Table 5: Besides the same criticism about the lack of statistics, what is the stress applied in Figure 9 a) and b)? What is the temperature applied in Figure 9 c) and d)? How is creep life calculated? Especially at high temperatures vs lower temperatures.
Response: Thank you for your comment. (1) It is very difficult to measure multiple repetitions due to their too long test period, we have disclosed the estimated errors on the presented values in the Table 5.
(2) The stress applied in Figure 9(a) and (b) is 12 MPa, and the temperature applied in Figure 9(c) and (d) is 25℃. We have revised this part, please see Lines 332-333 and related captions of Figure 9.
(3) Creep life corresponds to the time when the solder alloy fractures. After the creep tests, we can read the data form the creep time vs. strain curves as automatically recorded by dynamic mechanical analyzer (DMA).
Comment 8:
On Line 405 "highest strength ~64.8 MPa that is approximately 132% higher than that of SAC105". This is strange. 64.8/22.7 = 285% in my calculation.
Response: We are so sorry for our mistake. In the manuscript, we wrote “Higher than SAC105”, which is compared with SAC105, so our calculation is (64.8-22.7)/22.7=185%.
We would like to thank you again for your valuable and constructive comments.
Sincerely yours,
Xilei Bian

Reviewer 3 Report
The paper provides a very good overview of the possibilities of new alloys that make it possible to reduce the Ag content. Many investigations are presented and a large number of publications are cited. In addition to mechanical investigations, microstructure investigations are also used.
There are still some questions that have not been solved yet, but they would go beyond the scope of this paper.
Remarks:
Table 1 should not be spitted on two pages.
The lines 369-70 are not understandable written.
Lines 383-385 should include that lattice diffusion is the creep controlling mechanism.
The language of a few sentences should be revised.
Author Response
Dear reviewer,
We highly appreciate the time and efforts you have dedicated to providing valuable feedback. We are sorry that some presentations are still not quite clear and confused in the last manuscript. We have carefully addressed the issues raised by the reviewers, point by point. All changes in the revised manuscript are highlighted by red color. We believe your comments and suggestions have significantly improved the quality of our manuscript and made it publishable.
Comment 1:
The paper provides a very good overview of the possibilities of new alloys that make it possible to reduce the Ag content. Many investigations are presented and a large number of publications are cited. In addition to mechanical investigations, microstructure investigations are also used.
There are still some questions that have not been solved yet, but they would go beyond the scope of this paper.
Response: Thank you very much for your positive comments. We also would like to thank you for your interest in reading our manuscript, which greatly encourages us.
Comment 2:
Table 1 should not be spitted on two pages.
Response: Thank you very much for your suggestion. We have modified Table 1 as well as Table 5 that presented on one page. Please see the revised manuscript.
Comment 3:
The lines 369-70 are not understandable written.
Lines 383-385 should include that lattice diffusion is the creep controlling mechanism.
Response: Thank you very much the reviewer’s comment. We have revised the sentences. Please see the revised manuscript.
We would like to thank you again for your valuable and constructive comments.
Sincerely yours,
Xilei Bian

Reviewer 4 Report
Replacing the solder when welding material containing lead that is harmful to humans is an important issue. This article assumes the development of solder based on aluminum, tin, silver, copper and nickel additives. As a result, the authors find materials with high welding reliability and low melting points. The authors showed that a decrease in the concentration of silver in the solder can lead to degradation of the mechanical properties of the system. The paper shows the results obtained in specific methods, there is no generalization and comparison with the studies of other authors. Therefore, I believe that the article can be accepted for publication only if it is substantially revised.
Author Response
Dear reviewer,
We highly appreciate the time and efforts you have dedicated to providing valuable feedback. We have carefully addressed the issues raised by the reviewers and give explanations. We believe your comments and suggestions have significantly improved the quality of our manuscript and made it publishable.
Comment 1:
Replacing the solder when welding material containing lead that is harmful to humans is an important issue. This article assumes the development of solder based on aluminum, tin, silver, copper and nickel additives. As a result, the authors find materials with high welding
reliability and low melting points. The authors showed that a decrease in the concentration of silver in the solder can lead to degradation of the mechanical properties of the system. The paper shows the results obtained in specific methods, there is no generalization and comparison with the studies of other authors. Therefore, I believe that the article can be accepted for publication only if it is substantially revised.
Response: Thank you very much the reviewer’s comment.
On one hand, “This article assumes the development of solder based on aluminum, tin, silver, copper and nickel additives.” This comment seems not relate to our study. We didn’t study the addition of “aluminum”. On the other hand, the content of Ag is not the focus of our current study, we just choose SAC105 as the model material. The lines 39-43 mentions that:“Sn-1.0 Wt.% Ag-0.5Wt.% Cu (SAC105) was considered as promising candidate because of its good drop failure resistance in electronic interconnects, however, SAC105 still exists some questions need to solve, such as low strength, poor creep resistance, and poor wettability”. So, in this paper, we mainly research the effects and mechanism of microalloying by Sb、In、Ni and Bi elements on improving the mechanical properties, wettability, thermal properties and creep resistance of SAC105. Rather than focused on the study: “ a decrease in the concentration of silver in the solder can lead to degradation of the mechanical properties of the system.” After tensile tests, microstructure characterization, DSC tests, wettability tests and creep tests, we give strong supporting evidence and discuss the effects and mechanism of Sb, In, Ni, and Bi on microstructure, mechanical properties, melting point, wettability, and creep resistance of SAC solder alloys. In the end of the paper, the lines 400-431, we give five conclusions and fully explained the effects and mechanism of microalloying. It's worth noting that in this paper, we exploit a new solder by the addition of Sb, In, and Bi elements simultaneously, SAC105-2Sb-4.4In-0.3Bi owns high strength, good elongation and wettability, low melting temperature and good creep resistance. This proved that the performances of SAC105 can be much improved by four elements (Sb, In, Ni, and Bi), the comparison with SAC105 were listed in part 3 (Results and discussion). The lines 73-76, we point out that “Even though some studies have focused on the effects of Sb, In, Ni, and Bi on the microstructure and properties of SAC alloys, most of them solely took the effect of adding a single or two alloying elements into account. However, the simultaneous impact of additions of Sb, In, Ni, and Bi on the microstructure and properties of SAC105 alloys is still missing.” Then the additions of Sb, In, Ni, and Bi simultaneously and study the impact of SAC105 is the Innovation of our paper.
We would like to thank you again for your valuable and constructive comments.
Sincerely yours,
Xilei Bian
